# Unveiling the RKIP and EGFR Inverse Relationship in Solid Tumors: A Case Study in Cervical Cancer

**DOI:** 10.3390/cancers16122182

**Published:** 2024-06-10

**Authors:** Diana Cardoso-Carneiro, Joana Pinheiro, Patrícia Fontão, Rosete Nogueira, Maria Gabriela-Freitas, Ana Raquel-Cunha, Adriana Mendes, Adhemar Longatto-Filho, Fábio Marques, Marise A. R. Moreira, Rui M. Reis, Olga Martinho

**Affiliations:** 1Life and Health Sciences Research Institute (ICVS), Health Sciences School, University of Minho, 4710-057 Braga, Portugal; id7787@alunos.uminho.pt (D.C.-C.); id9964@alunos.uminho.pt (J.P.); id9967@alunos.uminho.pt (P.F.); rosete.nogueira@med.uminho.pt (R.N.); pg36885@alunos.uminho.pt (M.G.-F.); id8805@alunos.uminho.pt (A.R.-C.); id9003@alunos.uminho.pt (A.M.); longatto@med.uminho.pt (A.L.-F.); rreis@med.uminho.pt (R.M.R.); 2ICVS/3B’s—PT Government Associate Laboratory, 4710-057 Braga, Portugal; 3CGC Genetics/Centro de Genética Clínica, Unilabs-Laboratory of Pathology, 4000-432 Porto, Portugal; 4Medical Laboratory of Medical Investigation (LIM), Department of Pathology, Medical School, University of São Paulo, São Paulo 01246-903, SP, Brazil; 5Molecular Oncology Research Center (CPOM), Barretos Cancer Hospital, Barretos 14784-400, SP, Brazil; 6Department of Pathology, School of Medicine, Federal University of Goiás, Goiás 74605-050, GO, Brazil; fabim12@gmail.com (F.M.); marisemoreira7@gmail.com (M.A.R.M.)

**Keywords:** RKIP, HER, EGFR, cervical cancer

## Abstract

**Simple Summary:**

For the first time, we hypothesized an inverse correlation and almost mutual exclusivity between RKIP and EGFR expression in different solid tumors. In silico, cervical cancer was the third tumor type in which this inverse correlation was strong, a result that we validated in a series of 202 patient samples. Importantly, we highlight the importance of this axis by demonstrating its strong association with patient prognosis. Using in vitro functional assays, we showed that RKIP can control EGFR mRNA expression as well as its phosphorylation status, while EGFR activation can also interfere with RKIP protein expression levels. Therefore, the discovery of this novel putative RKIP/EGFR loop opens the door for several interesting future studies in many other tumor types, mainly those in which EGFR has a crucial oncogenic role.

**Abstract:**

Raf Kinase Inhibitor Protein (RKIP) is recognized as a bona fide tumor suppressor gene, and its diminished expression or loss is associated with the progression and poor prognosis of various solid tumors. It exerts multifaceted roles in carcinogenesis by modulating diverse intracellular signaling pathways, including those governed by HER receptors such as MAPK. Given the significance of HER receptor overexpression in numerous tumor types, we investigated the potential oncogenic relationship between RKIP and HER receptors in solid tumors. Through a comprehensive in silico analysis of 30 TCGA PanCancer Atlas studies encompassing solid tumors (10,719 samples), we uncovered compelling evidence of an inverse correlation between RKIP and EGFR expression in solid tumors observed in 25 out of 30 studies. Conversely, a predominantly positive association was noted for the other HER receptors (ERBB2, ERBB3, and ERBB4). In particular, cervical cancer (CC) emerged as a tumor type exhibiting a robust inverse association between RKIP and EGFR expression, a finding that was further validated in a cohort of 202 patient samples. Subsequent in vitro experiments involving pharmacological and genetic modulation of EGFR and RKIP showed that RKIP depletion led to significant upregulation of EGFR mRNA levels and induction of EGFR phosphorylation. Conversely, EGFR overactivation decreased RKIP expression in CC cell lines. Additionally, we identified a common molecular signature among patients depicting low RKIP and high EGFR expression and demonstrated the prognostic value of this inverse correlation in CC patients. In conclusion, our findings reveal an inverse association between RKIP and EGFR expression across various solid tumors, shedding new light on the underlying molecular mechanisms contributing to the aggressive phenotype associated with RKIP and EGFR in cervical cancer.

## 1. Introduction

Cervical cancer ranks as the fourth most common cancer among women worldwide, contributing to approximately 8% of total cancer-related deaths [1]. The primary etiological risk factor is infection by high-risk human papillomavirus (HPV) [2]. Treatment strategies depend heavily on the disease stage. For women with low-risk, early-stage disease, conservative, fertility-preserving surgical procedures are the standard of care. For locally advanced disease, chemo-radiotherapy is the typical treatment. Unfortunately, women with recurrent or metastatic disease have limited treatment options, resulting in a poor overall prognosis. Platinum-based chemotherapy is often used as palliative care [2,3,4].

The introduction of bevacizumab, a monoclonal antibody targeting Vascular Endothelial Growth Factor A (VEGF-A), has extended the overall survival of this group of women beyond 12 months [5]. More recently, the Food and Drug Administration (FDA) approved immunotherapy with pembrolizumab, which offers significantly longer progression-free and overall survival compared to placebo among patients with persistent, recurrent, or metastatic cervical cancer who also receive chemotherapy, with or without bevacizumab [6]. However, precision medicine treatment options remain limited and have not significantly improved cervical cancer patient survival [7]. In our quest to identify molecularly targeted therapies that prolong life without adding toxicities, we have recently suggested using Human Epidermal Growth Factor Receptor (HER) inhibitors as a potentially effective treatment for HER-positive cervical cancer patients [8].

HER, also known as ErBb, is part of the RTK (Receptor Tyrosine Kinase) family, which includes HER1 or EGFR (Epidermal Growth Factor Receptor), HER2, HER3, and HER4. These receptors interact with 13 different but structurally related growth factors, such as EGF; however, no soluble ligand has been identified for HER2 [9]. Amplification, overexpression, and co-expression of HER family receptors have been implicated in the genesis or progression of various malignancies [9,10]. Notably, HER inhibitors are among the most successful examples of targeted cancer therapies to date [11,12,13]. In cervical cancer, our comprehensive analysis of HER family receptor alterations revealed significant HER protein overexpression in many cases, particularly HER2, which emerged as an independent prognostic marker for these patients [8]. We also demonstrated that HER inhibitors effectively reduce cervical cancer aggressiveness, suggesting their use alongside glycolysis blockers as a potentially effective treatment for HER-positive cervical cancer patients [8]. Although the concept of oncogene addiction has shown initially impressive clinical results in other tumor types [14], it has been observed that some patients benefit more from anti-RTK therapy than others in clinical practice [15,16]. Thus, a critical question remains: how can we identify which cervical cancer patients are most likely to benefit from this therapy?

An essential effector cascade required for most RTK functions is the Mitogen-Activated Protein Kinase (MAPK) pathway, which can be disrupted in human cancers by activating mutations in *BRAF* and *KRAS* oncogenes or by downregulation of Raf Kinase Inhibitor Protein (RKIP) [17,18]. MAPK pathway alterations are the most important modulators and predictors of anti-RTK therapy response; for example, the presence of *KRAS* mutations is a negative predictor of response to EGFR inhibitors in colon cancer patients [15]. In cervical cancer, *KRAS* or *BRAF* mutations occur at a very low-frequency [19]. In contrast, we found that RKIP expression was significantly lost during the malignant progression of cervical cancer, being highly expressed in non-neoplastic tissues and expressed at very low levels in invasive carcinomas. In addition, we also demonstrated in vitro and in vivo that the loss of RKIP expression could be one of the factors behind the aggressiveness, malignant progression, and chemotherapy resistance of these tumors [20].

RKIP or PEBP1 is an endogenous inhibitor of the Extracellular Regulated Kinase (ERK)/MAPK pathway [21,22], but several studies have shown lately that RKIP is a multifunctional protein in carcinogenesis, which through the modulation of different intracellular signaling pathways, can control different biological processes [22,23]. RKIP is considered a tumor suppressor in cancer, and its loss or reduced expression is associated with malignancy and prognosis in many types of metastatic and aggressive cancers, as has already been described by our and other groups (reviewed in [18]).

Taken together, our previous findings led us to consider RKIP downregulation and HER receptor overexpression as critical molecular alterations in the carcinogenic process of cervical cancer [8,20]. Adding to this the well-established role of RKIP as a modulator of signaling pathways [18], we wondered whether RKIP and HER receptors are oncogenically related and if RKIP could be a putative regulator of HER targeted therapy response in cervical cancer.

## 2. Materials and Methods

### 2.1. In Silico Analysis

The cBioPortal for Cancer Genomics (http://www.cbioportal.org, last accessed on 14 April 2024) is a repository of cancer genomics datasets that were used to analyze RKIP and HER receptor alterations, firstly in 30 different tumor types using the TCGA PanCancer Atlas studies, which comprise 10,719 samples. Secondly, and specifically for cervical cancer, in the present study, the Cervical Squamous Cell Carcinoma and Endocervical Adenocarcinoma (TCGA, Firehose Legacy) database with 310 samples currently available from 308 patients was used. The Cancer Cell Line Encyclopedia (Broad 2019) database is a repository of the genetic basis of different cancer cell lines, comprising 1739 cell lines. According to the TCGA guidelines (http://cancergenome.nih.gov/publications/publicationguidelines, last accessed on 14 April 2024), this dataset has no limitations or restrictions [24,25].

Regarding patient data analysis, alterations in RKIP and HER receptor expression at the mRNA and protein levels were analyzed across the set of samples available and summarized in the format of expression heatmaps, plots, co-expression plots, or expression plots generated in cBioPortal. For expression analysis, we assessed log-transformed mRNA expression z-scores against the expression distribution of diploid samples (RNA Seq V2 RSEM) and protein expression z-score (RPPA) data. Significant alterations in mRNA and protein expression were identified using a z-score threshold of ±2, unless otherwise specified. Spearman correlations were directly calculated using cBioPortal to determine correlations between the genes of interest.

For survival analyses and to determine the enriched proteins in the group of patients that had low RKIP expression concomitant with high EGFR expression, we arbitrarily lowered the z-score to 0 to increase the sample size. Kaplan–Meier survival analysis was performed using cBioPortal, considering patient tumor samples with both low and high RKIP-low (PEBP1: EXP < 0 and EGFR: EXP > 0; N = 99) or both RKIP-high and EGFR-low (PEBP1: EXP > 0 and EGFR: EXP < 0; N = 107) mRNA expression and patients with only RKIP-low (PEBP1: EXP < 0; N = 71) or RKIP-high (PEBP1: EXP > 0; N = 26) expression. Concerning the analysis to determine the enriched proteins in the group of patients that had low RKIP expression concomitantly with high EGFR protein expression, we divided the patient tumor samples into four groups: RKIP-low mRNA expression and EGFR-high protein expression (PEBP1: EXP < 0 and EGFR: PROT > 0; N = 48); RKIP-low mRNA expression and EGFR-low protein expression (PEBP1: EXP < 0 and EGFR: PROT < 0; N = 43); RKIP-high mRNA expression and EGFR-high protein expression (PEBP1: EXP > 0 and EGFR: PROT > 0; N = 30); and RKIP-high mRNA expression and EGFR-low protein expression (PEBP1: EXP > 0 and EGFR: PROT < 0; N = 50). By comparing these 4 groups, the significantly enriched proteins were retrieved from cBioPortal for functional analysis. We conducted functional gene-set enrichment analysis of the previously mentioned highly expressed proteins using the graphical gene-set enrichment tool in the ShinyGO V0.80 software (http://bioinformatics.sdstate.edu/go74/, last accessed on 14 April 2024) [26]. For identifying enriched pathways, the KEGG category was selected, with an adjusted *p*-value (FDR) cut-off of 0.05 set as the significance threshold. To analyze the functional protein association network, we utilized STRING v11.5 (https://string-db.org/, last accessed on 14 April 2024) [27], which helped assess the interactions among the enriched proteins. A protein–protein interaction (PPI) enrichment *p*-value below 0.05 was considered indicative of significant network interaction enrichment.

The Gene Expression Profiling Interactive Analysis (GEPIA) tool (http://gepia2.cancer-pku.cn, last accessed on 14 April 2024) was used to assess RKIP and EGFR mRNA expression levels and conduct survival analysis using TCGA data across 30 cancer types. Kaplan–Meier survival analysis was performed using the median as the expression threshold to divide the cohorts into high- and low-expression groups. Hazard ratios (HRs) were calculated using a Cox proportional hazards model. HR, heatmap, and Kaplan–Meier curves were generated directly from GEPIA2. A log-rank test with a significance threshold of <0.05 was applied for statistical analysis.

### 2.2. Tissue Samples

In this study, we analyzed the same series previously used by our group to study HER receptor expression [8]. A total of 229 cervical cancer (CC) tissues, including 202 adenocarcinomas, were used. Paraffin-embedded samples included in tissue microarrays (TMAs) were obtained from the files of the School of Medicine, Federal University of Goiás (Goiânia, Goias State, Brazil). All histopathological diagnoses were reviewed and classified according to the WHO classification [28]. The patients, all of Brazilian origin, had a mean age of 47 years, with an age range of 22–84 years.

### 2.3. Immunohistochemistry Analysis (IHC) for RKIP

Histological slides with 4 µm thick tissue sections were subjected to immunohistochemical analysis based on the streptavidin–biotin–peroxidase principle (UltraVision Large Volume Detection System Anti-Polyvalent, HRP; LabVision Corporation, Fremont, CA, USA), as previously described [20]. Briefly, deparaffinized and rehydrated slides were subjected to heat-induced antigen retrieval for 20 min at 98 °C using 10 mM citrate buffer (pH 6.0). Following the blocking step, the slides were incubated with RKIP antibody (RKIP 1:1200, 07-137, EMD Millipore, Burlington, MA, USA) overnight at 4 °C. The secondary biotinylated goat anti-polyvalent antibody was then applied for 10 min, followed by incubation with the streptavidin–peroxidase complex. The immune reaction was visualized using 3,3′-diaminobenzidine (DAB) as a chromogen, and all sections were counterstained with Gill-2 hematoxylin. A positive control was included in the immunoreaction.

Tissue immunostaining was scored by an experienced professional, considering both extension and intensity. The scoring for immunoreactive extension was as follows: score 0, 0% immunoreactive cells; score 1, <30% immunoreactive cells; score 2, 30–50% immunoreactive cells; and score 3, >50% immunoreactive cells. For intensity, the scores were as follows: 0, negative; 1, weak; 2, intermediate; and 3, strong. The final score was determined by summing these two semi-quantitative scores, with a total score between 0 and 2 classified as negative, between 3 and 4 as moderately positive, and between 5 and 6 as strongly positive. For statistical analysis, a final score greater than 2 was considered positive for RKIP expression, and a score greater than 4 indicated overexpression of RKIP and HER receptors [8].

### 2.4. Cell Lines and Culture Conditions

Five cervical cancer (CC) cell lines were used in this study: HeLa, SiHa, SW756, C-33A, and CaSki. The HeLa cell line was obtained from ATCC, while the SiHa, C-33A, CaSki, and SW756 cell lines were kindly provided by Dr. Luisa Villa (Instituto Nacional de Ciência e Tecnologia do HPV, Brazil). These cell lines were cultured and maintained at 37 °C with 5% CO_2_ in Dulbecco’s Modified Eagle’s medium (DMEM 1x, High Glucose; Gibco, Invitrogen, Waltham, MA, USA) supplemented with 10% Fetal Bovine Serum (FBS; Gibco, Invitrogen) and 1% penicillin/streptomycin solution (Gibco, Invitrogen, Waltham, MA, USA).

In vitro RKIP knockout (KO) was achieved using a CRISPR/Cas9-based kit (Santa Cruz Biotechnology, Dallas, TX, USA). Cells were transfected with a control plasmid (HDR Plasmid, Sc-401270-HDR-2) to generate control cells (CTR), and with both the control and Cas9 plasmid (CRISPR/Cas9 KO Plasmid, sc-401270) to create RKIP KO cell lines. Stably transfected cells were selected with 2 µg/mL puromycin. RKIP overexpression (OE) was induced using a pcDNA3 empty vector as control and a pcDNA3 vector containing the full cDNA of RKIP, kindly provided by Dr. Evan Keller from the University of Michigan (Ann Arbor, MI, USA). Stably transfected cells were selected with 2000 µg/mL G418. For both knockout and overexpression approaches, transfections were performed using FUGENE HD reagent (Roche, Porto, Portugal), following the manufacturer’s instructions.

### 2.5. Western Blot Analysis

Cells were plated in a 6-well plate at a density of 1 × 10^6^ cells per well and allowed to adhere for at least 24 h. Subsequently, the cells were serum-starved for 2 h and, in some cases, incubated with the respective drugs at a concentration of 2.5 µM for 2 h, as previously described [8]. When necessary, cells were stimulated with 10 ng/mL of EGF for 15 min or more before the end of the incubation period. The cells were then washed and scraped in cold PBS and lysed in buffer, as previously described [8].

Western blotting was performed using standard 12% SDS-PAGE, with 50 µg of protein loaded per lane. All antibodies were used according to the manufacturer’s recommendations as previously described [8]. α-Tubulin (Santa Cruz Biotechnology, Dallas, Texas, USA, dilution 1:1000) served as a loading control when necessary. Blot detection was carried out by chemiluminescence using WesternBright Sirius HRP substrate (Advansta, San Jose, CA, USA) and the Sapphire Biomolecular Imager (Azure Biosystems, Dublin, CA, USA). Quantification of Western blot results was performed through band densitometry analysis using ImageJ software version 1.8. Relative protein expression results are presented as the ratio of target proteins to α-tubulin. The results are shown as the mean values obtained from at least two independent assays, quantified independently by three researchers.

### 2.6. Quantitative Real-Time Analysis

RNA was extracted from the cell lines using TRIzol reagent (Invitrogen) according to the manufacturer’s instructions. Cells were lysed with TRIzol, and DNA and RNA were separated using chloroform-based phase separation. RNA precipitation was performed using isopropanol 100%, and then it was washed with ethanol 75%. Reverse transcription of 1 μg of RNA into cDNA was performed using Xpert cDNA Synthesis Kit (GRiSP Research Solutions), according to the manufacturer’s instructions.

The mRNA expression levels of RKIP and EGFR were assessed by qRT-PCR assay. β-actin was used as reference gene. For qRT-PCR, the SsoFast™ EvaGreen^®^ Supermix (Bio-Rad Laboratories, Hercules, CA, USA) was used. The PCR reaction consisted of 1 μL of cDNA, 5 μL of EvaGreen Supermix, 0.3 μM of each primer, and RNase-free H_2_O. PCR mix was run on a Thermal cycler CFX96 (BioRad Laboratories, Hercules, CA, USA), and PCR conditions were as follows: 95 °C for 30 s; 40 cycles of 95 °C for 5 s for denaturation; and 56 °C (near the melting temperature of the primers) for 5 s for annealing and extension; lastly, the temperature was gradually increased in 0.5 °C from 65 °C to 95 °C for 5 s. PCR mixture without the cDNA template was used as negative control. The primers used were the following: RKIP Forward 5′-GACATCAGCAGTGGCACAGT-3′ and RKIP Reverse 5′-GTCACACTTTAGCGGCCTGT-3′; EGFR Forward 5′-TTTGGGAACGGACTGGTTTA-3′ and EGFR Revers 5′-GCCTTGACTGAGGAAGAGCA-3′, as described in [29]; β-actin Forward 5′-GGACTTCGAGCAAGAGATGG-3′ and β-actin Reverse 5′-AGCACTGTGTTGGCGTACAG-3′, as described in [30].

### 2.7. Statistical Analysis

The chi-squared test was used to examine the correlations between protein expression and clinical data from patients. Kaplan–Meier method was used to calculate cumulative survival probabilities, and differences in survival rates were evaluated using the log-rank test. Statistical analysis was conducted using SPSS software version 28.0.

For in vitro assays, individual comparisons among the various studied conditions were conducted using Student’s *t*-test, whereas differences between groups were assessed using two-way analysis of variance (ANOVA). Statistical analyses were performed using the GraphPad Prism version 10. The significance level for all statistical analyses was set at *p* < 0.05.

## 3. Results

### 3.1. Correlation between RKIP and HER Receptor Expression in Solid Tumors

RKIP is a multifunctional protein in carcinogenesis, which modulates different signaling pathways, including the ones regulated by HER receptors [22]. Thus, we wondered whether HER receptors and RKIP are co-expressed in cancer. To do so, we referred to The Cancer Genome Atlas (TCGA) data, available at cBioPortal (www.cbioportal.org, last accessed on 14 April 2024), and started by evaluating the 30 TCGA PanCancer Atlas studies related to solid tumors (10,719 samples) for RKIP (*PEBP1* gene) and HER receptor (EGFR, ERBB2, ERBB3, and ERBB4) expression. At the mRNA level, we found that *PEBP1* is mutually exclusively expressed with EGFR and co-expressed with the other HER receptors (Table 1). Accordingly, using co-expression plots, we observed that *PEBP1* expression was significantly inversely correlated with *EGFR* expression (Spearman coefficient = −0.32) but not with the other HER receptors (Table 1). Knowing this, we determined the Spearman correlation coefficients for each of the individual 30 PanCancer studies (Appendix A). We corroborated the finding of a statistically significant and strong inverse correlation between *RKIP* and *EGFR* mRNA expression in almost all tumor types (25 out of 30), whereas for the other HER receptors, the correlation observed was mostly positive (Figure 1A).

The TCGA database has no information on RKIP or HER4 protein expression from the reverse-phase protein array (RPPA), hampering direct protein–protein correlations in clinical samples. Even so, we estimated the correlation levels between RKIP mRNA and HER receptor protein expression for all datasets (Appendix A). Even with lower statistical significance than at the mRNA level, the tendency for RKIP to be inversely correlated with EGFR protein was maintained for a high number of tumor types, whereas for HER2 and HER3 proteins, the correlations were mainly positive (Appendix A). Focusing on EGFR protein correlations and adding data regarding phosphorylated forms of the protein (Appendix A), strong statistically significant inverse correlations were maintained in nine tumor types (LGG, BRCA, STAD, LUAD, LUSC, CESC, OV, BLCA, THCA); with five of them (BRCA, STAD, LUSC, CESC, and BLCA), the correlation was also maintained at the phosphorylated level (Figure 1B).

In conclusion, we provided strong and consistent evidence that RKIP is inversely correlated with EGFR in solid tumors.

### 3.2. Clinical Impact of the RKIP/EGFR Negative Feedback Loop in Cervical Cancer

From the above in silico analysis, we observed that cervical squamous cell carcinoma (SCC) is the third tumor type in which RKIP is most significantly inversely correlated with EGFR at the mRNA level (Figure 1A) and the second at the protein level when considering total and phosphorylated proteins (Figure 1B). Moreover, referring to survival maps from GEPIA2 (Gene Expression Profiling and Interactive Analyses vision 2), CESC was revealed to be a tumor type for which low PEBP1 expression and high EGFR intensity were associated with a worse survival rate of patients, indicating a putative clinical significance for this RKIP/EGFR axis (Appendix A).

To further dissect the translational impact of RKIP and EGFR negative regulation in CC, we performed immunohistochemistry (IHC) analysis of RKIP in a series of CC human samples, which we previously characterized for HER expression (Figure 2) [8].

In our series of 202 adenocarcinoma samples, RKIP was found to be expressed not only in the nucleus and cytoplasm of tumor cells but also in the stroma (Figure 2A). Overall, RKIP was overexpressed in 53.9% (113/202) of the tumors, moderately expressed in 32.7% (66/202), and in 11.4% (23/202) of the tumors was found to be negative. Crossing the data for RKIP and HER immunohistochemical expression, we corroborated the finding of a significant negative association between RKIP and EGFR overexpression (*p* = 0.029). Excepting HER3, which was positively associated (*p* = 0.018), no other HER receptors were associated with RKIP protein (Figure 2B and Appendix A). Concerning associations with clinicopathological data (Appendix A), low RKIP expression was significantly associated with the presence of metastases and poor prognosis (*p* = 0.026) in cervical adenocarcinomas (Figure 2C and Appendix A). In this series of CC samples, only seven patients overexpressed EGFR and only one of them co-expressed RKIP (Appendix A), which hampered the stratification of the Kaplan–Meier curves according to EGFR expression levels. In fact, in our previous study, we showed that EGFR was differentially expressed between CC histological types and was less expressed in the adenocarcinoma subtype [8]. Thus, by re-accessing the CESC database (TCGA, Firehose Legacy) available at cBioPortal, composed of 310 samples derived from 308 patients, we observed that the impact of RKIP expression on patient survival was stronger when combined with the expression levels of EGFR; that is, patients with concomitantly low RKIP and high EGFR expression presented a poorer outcome than patients with high RKIP and low EGFR expression or even than patients with low RKIP alone (Figure 2D).

In conclusion, we demonstrated that RKIP and EGFR are negatively correlated at the protein level in human CC samples and that this putative negative feedback loop can constitute an important prognostic marker for patients.

### 3.3. Oncogenic Interplay between RKIP and EGFR Cervical Cancer Models

To further explore this interplay, we used in vitro approaches to determine whether EGFR and RKIP are functionally related and can regulate each other in a negative feedback loop. First, using the Cancer Cell Line Encyclopedia (Broad, 2019) database available at cBioPortal, which comprises 1739 human cancer cell lines and contains information for both RKIP mRNA and protein expression, a statistically significant negative correlation between RKIP and EGFR was confirmed at both the mRNA and protein levels (Appendix A).

In the Cancer Cell Line Encyclopedia (Broad, 2019) database, there are no data for CC cell lines; therefore, we began by characterizing the basal expression levels of the two proteins of interest, both at the protein and mRNA levels, in a panel of five CC cell lines (Figure 3). By Western blot analysis, it was possible to notice that EGFR and RKIP proteins seem inversely correlated because all cell lines expressed higher levels of RKIP than EGFR, mainly its phosphorylated form, except the CaSki cell line, which expresses higher levels of EGFR than RKIP, being the cell line that showed the lowest levels of RKIP (Figure 3A). Quantification of RKIP and EGFR mRNA levels revealed the same variance and tendency as Western blotting (Figure 3B).

To determine the influence of EGFR signaling on RKIP expression, we started by inducing EGFR phosphorylation by stimulation with EGF. Except for the SW756 and CaSki cell lines, RKIP protein expression significantly decreased upon EGF treatment (Figure 3C,D). Curiously, SW756 and CaSki are the cell lines with the highest EGFR phosphorylation levels at basal conditions, meaning that they are hypothetically under EGFR control, and the quantity or time of EGF treatment was not enough to impact RKIP expression. To test this hypothesis, we treated the SW756 cell line with the same dose of EGF (10 ng/mL), but over time, starting with the standard 15 min and ending before the EGFR turnover of 35 min. In this experiment, we progressively activated EGFR for a longer period, which allowed us to observe a concomitant decrease in RKIP expression, even in an overactivated EGFR cell line (Figure 3E).

To further validate these findings, we inhibited EGFR phosphorylation using a specific inhibitor, erlotinib. For this purpose, we selected the highest EGFR-expressing cell lines (SW756 and CaSki) and another that responded to EGF (Siha). Thus, EGFR tyrosine kinase inhibition increased RKIP expression levels in the SiHa cell line but not in SW756 and CaSki cells (Figure 3F), as observed in the EGF experiments (Figure 3C). We observed that EGFR phosphorylation was not completely abolished in these two cell lines, suggesting that the remaining phosphorylated protein was sufficient to maintain RKIP expression (Figure 3F). However, the effect of EGFR signaling on RKIP expression seemed to be dependent on the experimental conditions, as shown in Figure 3E. When we exposed the cells to increasing concentrations of the EGFR inhibitor, RKIP increased concomitantly (Appendix A). Thus, the putative regulatory role of EGFR under RKIP expression, at least in vitro, seems to be post-translational and dependent on EGFR activation status (phosphorylation), since EGFR activity disturbance had no impact on its total levels, nor on EGFR and RKIP mRNA expression (Appendix A).

To determine whether RKIP can regulate EGFR, we genetically modulated its expression by in vitro knockout (KO) and plasmid overexpression (RKIP+) in SiHa and SW756 cell lines (Figure 4). It was observed that RKIP KO significantly increased EGFR at the protein level, which was more perceptible in its phosphorylated form, only in the SiHa cell line (Figure 4A,B). At the mRNA level, *EGFR* increased significantly in both cell lines upon RKIP KO (Figure 4C,D). In contrast, overexpression of RKIP resulted in no significant changes in *EGFR* mRNA levels, but a significant decrease in its activation levels was observed in both cell lines (Figure 4A,B).

In conclusion, we provide functional evidence that RKIP depletion leads to a significant increase in *EGFR* mRNA levels and induces its phosphorylation. This effect is particularly pronounced in cell lines that exhibit low basal activation levels of EGFR and lack MAPK alterations (Siha). In contrast, RKIP overexpression resulted in a slight decrease in EGFR activation, without a notable impact on its mRNA levels. EGFR activation affects RKIP expression only at the protein level.

### 3.4. Molecular Signature of RKIP and EGFR Negative Feedback Loop in Cervical Cancer

To identify other proteins that were concomitantly overexpressed in a subset of patients with low RKIP and high EGFR levels, we conducted an analysis using the TCGA CESC dataset followed by enrichment analysis. In addition to the anticipated presence of EGFR, our investigation revealed 14 other proteins that showed significant alterations in this specific patient subset (Figure 5A and Table 2). Thereafter, we performed functional enrichment and protein association network analyses to determine the enriched pathways present in this set of proteins that were altered in these patients. With this analysis, we concluded that the set of overexpressed proteins presented several enriched pathways (Figure 5C) and that the network presented a high level of protein–protein interaction (Figure 5B). It is noteworthy that of all the pathways that appeared, the ErbB signaling pathway had a significant enrichment value. Furthermore, the proteins highly expressed in these patients were involved in several distinct processes such as HIF-1 signaling, human papillomavirus infection, and EGFR tyrosine kinase inhibitor resistance, among others (Figure 5C). Interestingly, some of the proteins were found to be overexpressed in their phosphorylated status, such as EGFR and PDK1 proteins (Table 2), suggesting an effective concomitant involvement of EGFR and RKIP in signaling modulation.

## 4. Discussion

Cervical cancer (CC) is a major malignant disease affecting women’s health worldwide [1]. Unfortunately, women with recurrent or metastatic disease have a median overall survival of only 8–12 months [8]. To unveil new molecular players involved in CC carcinogenesis, our group previously identified RKIP downregulation and HER overexpression as key events in this process [8,20] and suggested HER as a targetable protein to treat HER-positive CC patients [8]. In this study, we aimed to elucidate the potential oncogenic relationship between these two key molecules by investigating their regulatory interactions and their consequential impact on patient prognosis.

To address the putative association between RKIP and HERs, we performed a comprehensive in silico analysis of RKIP and HER co-expression in different solid tumors. For all 30 tumor types evaluated, we found a strong negative correlation between *RKIP* and *EGFR* mRNA levels. In contrast, the correlation of RKIP with *HER2*, *HER3*, and *HER4* was mostly positive, indicating a unique relationship between RKIP and EGFR. In fact, our analyses unveiled a pattern of mutual exclusivity in the expression of RKIP and EGFR. This negative correlation between RKIP mRNA and EGFR was maintained at the protein level for nine tumor types, suggesting a major role of RKIP in modulating EGFR at the transcriptional level. However, RKIP mRNA appears to be correlated with phosphorylated EGFR, prompting us to further investigate the relationship between RKIP and EGFR activation in this study.

In our study model, CC, specifically the SCC subtype, in silico analysis showed strong correlations between RKIP mRNA and EGFR both at mRNA and protein levels, including phosphorylated EGFR. We further explored the putative effect of these molecules, both individually and in combination, on patient prognosis. First, using the GEPIA2 database to assess the individual value of these molecules and according to other reports in the literature for several solid tumors [18,31,32], we found that both molecules are significant prognostic biomarkers. Second, and most importantly, using our human series [8], we were able to validate the RKIP value as a prognostic marker by uncovering, for the first time, a significant association between poor prognosis and low RKIP expression in human CC samples, particularly for the adenocarcinoma subtype. In addition, similar to the findings of Hu, C.J. et al., we were able to associate low RKIP expression with the presence of metastases [33]. Using TCGA database, which predominantly consists of squamous cell carcinoma (SCC) subtype data, we have elucidated that the combined influence of RKIP and EGFR expression significantly impacts patient prognosis, surpassing the prognostic relevance of RKIP expression alone. Notably, our previous investigation of SCC samples failed to establish a direct association between RKIP and patient prognosis [20]. This finding underscores the critical importance of exploring this molecular axis to understand and predict patient outcomes.

Significantly, through our CC series analysis of adenocarcinomas [8], we successfully confirmed an inverse correlation between RKIP and EGFR at the protein level. In addition, we identified a positive association between RKIP and HER3 expression. Notably, RKIP protein expression was observed, wherein only 11% of tumors exhibited RKIP negativity. This percentage contrasts starkly with our previous findings, where 84% of tumors lacked RKIP expression [20], and with the observations of Hu, C.J. et al., who reported 55.2% negative cases [33]. Analyzing the histological subtypes that compose each series, it seems that a higher percentage of RKIP-negative cases is related to a higher number of SCCs represented in the series, suggesting that RKIP loss could be a more frequent event in SCC. Interestingly, some studies have reported increased expression of EGFR as a more frequent event in SCC [34,35], and it was clear in our previous work that EGFR was the least expressed HER in a series of adenocarcinomas analyzed, while HER3 showed a higher expression [8]. Altogether, these results demonstrate that RKIP expression is higher when EGFR is expressed at low levels, as is the case for adenocarcinomas, and that RKIP is positively correlated with HER3, which appears to be more expressed in this histological subtype. Therefore, we suggest a pattern of negative correlation, almost mutual exclusivity, between RKIP and EGFR as a common feature of CC.

To corroborate this significant RKIP/EGFR association found in silico and in the tumor specimens, we characterized a panel of SCC cell lines (SiHa, SW756, C-33A, CaSki) and adenocarcinomas (HeLa) for RKIP and EGFR expression at the protein and mRNA levels. Despite RKIP and EGFR expression, both at the protein and mRNA levels, being quite different across the cell line panel, cell lines with higher levels of RKIP were among those with lower levels of EGFR.

Following in silico evidence of an association between RKIP and EGFR activation, we observed a decrease in RKIP levels upon stimulation with EGF, specifically in cell lines with lower basal levels of EGFR activation. Importantly, we were also able to demonstrate that even in cells with higher basal levels of EGFR activation, the activation of EGFR for longer periods results in decreased RKIP expression, leading us to conjecture the impact of EGFR activation status on RKIP expression. After pharmacological inhibition of EGFR activation using erlotinib, we observed an increase in RKIP expression. Remarkably, this increase was proportional to the EGFR inactivation level. However, both RKIP and EGFR mRNA expression levels were maintained, pointing to post-transcriptional regulation of RKIP mediated exclusively by EGFR activation status.

Indeed, an increase in RKIP has already been observed despite erlotinib treatment in NSCLC cells [36]. Also, this RKIP upregulation mediated by erlotinib was pointed out as the mechanism behind the potent synergistic interaction between erlotinib and sorafenib described [36].

Going further, we addressed RKIP’s impact on EGFR expression and activation by manipulating RKIP expression in vitro. The impact of RKIP modulation on EGFR expression at the mRNA level was clear; however, at the protein level, RKIP modulation only affected EGFR activation and not total EGFR expression. Overall, these results highlight RKIP as an EGFR modulator at the transcriptional and post-transcriptional levels following RKIP depletion, contributing to higher levels of both EGFR mRNA and EGFR phosphorylation. Previous studies by Trakul, N. et al. described RKIP as a regulator of EGF signaling [37]. They demonstrated that EGF concentrations that are normally rate-limiting can induce a better response to EGF stimulation in H19-7 and 293T RKIP-depleted cells, specifically a robust ERK activation and DNA synthesis [37]. These observations support our results because at the same EGF concentration, RKIP-depleted cells showed increased EGFR activation, demonstrating an increased sensitivity to EGF. Interestingly, a recent study described a higher dependency on EGFR signaling in cervical SCC cell lines, specifically after blockage of Aurora Kinases A and B [38], proteins that were shown to be less active in RKIP-depleted cells [39,40]. After Aurora Kinase A and B inhibition, phosphorylated EGFR expression increased, and, similar to our results, the authors were not able to detect differences in EGFR protein levels by Western blot; however, they detected EGFR accumulation on the cell membrane by immunofluorescence [38]. In addition, the authors demonstrated that this enhanced EGFR activation contributes to an increased sensitivity to erlotinib treatment that is not observed in normal cells, which indicates that this high dependency on EGFR signaling is a cancer-specific event [38], similar to RKIP downregulation [18].

To gain a deeper understanding of the targets that are potentially modifiable within a subset of samples exhibiting an inverse correlation between RKIP mRNA expression and EGFR protein expression, we endeavored to delineate a molecular signature using computational methods. Given the well-established roles of RKIP and EGFR in regulating various signaling pathways, it was anticipated that the identified proteins would predominantly belong to signaling cascades, with some appearing in the phosphorylated state. Notably, we observed enrichment of phosphorylated EGFR (Y1068) in samples with low RKIP mRNA expression, suggesting that RKIP may not only influence EGFR mRNA and protein expression but also modulate EGFR activation.

Both RKIP and EGFR have individually been associated with more aggressive phenotypes in CC [8,20]. Herein, we unveil a novel interplay between these two proteins, highlighting the significance of understanding the EGFR/RKIP oncogenic relationship, which is pivotal for identifying the players involved and devising strategies for novel therapeutic approaches.

## 5. Conclusions

In this study, we report for the first time an in silico association between RKIP and EGFR in different solid tumors. Specifically, for cervical cancer, by in silico and tumor specimen analyses, we found the same strong negative correlation, almost mutual exclusivity, between RKIP and EGFR both at mRNA and protein levels. Importantly, we demonstrated the importance of this axis in patient prognosis by finding that the impact of RKIP expression on patient prognosis is stronger when associated with EGFR expression. Additionally, using in vitro approaches, we characterized this novel EGFR/RKIP loop, which we believe is related to the oncogenic role of these proteins already described in cervical cancer. We conclude that EGFR activation status could be an important factor in EGFR-dependent RKIP modulation, while RKIP-dependent EGFR modulation occurs with both EGFR and activated EGFR.

## Figures and Tables

**Figure 1 cancers-16-02182-f001:**
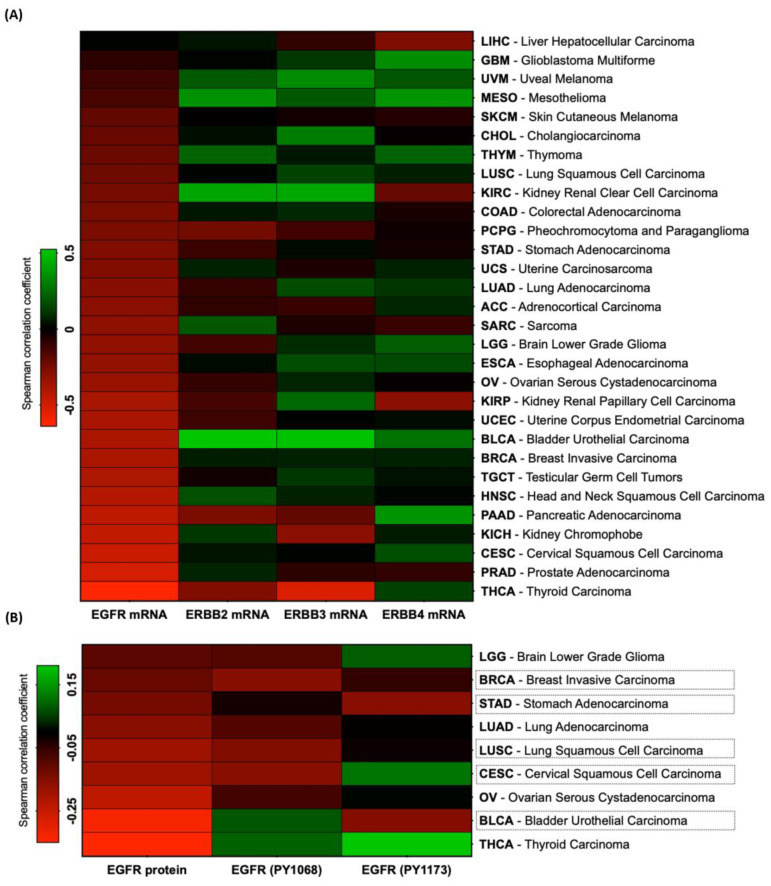
Correlation between RKIP and HER receptor expression in solid tumors—in silico analysis. Using TCGA data available at cBioPortal (www.cbioportal.org, last accessed on 14 Apri 2024), we analyzed the 30 TCGA PanCancer Altlas studies concerning solid tumors, which comprise 10,719 samples, and co-expression plots were generated to determine the correlation level between RKIP (PEBP1) and HER receptor (EGFR, ERBB2, ERBB3, ERBB4) expression. HER and PEBP1 mRNA expression data (RNA Seq V2 RSEM) were used relative to diploid samples, and EGFR protein (Reverse-phase protein array-RPPA). The correlation levels were assessed using Spearman’s correlation coefficient (*p*). (**A**) Individual Spearman’s correlation coefficients (Appendix A) regarding mRNA correlations are graphically represented in the heatmap. (**B**) Correlation levels between PEBP1 mRNA and EGFR protein, both total protein and the phosphorylated forms at tyrosine 1067 (PY1067) and tyrosine 1173 (PY1173). Spearman correlation coefficients are graphically represented as a heatmap for the 9 tumor types in each the inverse correlation with total protein was significant (Appendix A). BRCA, STAD, LUSC, CESC, and BLCA are the 5 datasets in which the correlation was maintained at the phosphorylated level. PEBP1 expression was classified as positively (*p* > 0; dark to green) or negatively (*p* < 0; dark to red) correlated with HER expression.

**Figure 2 cancers-16-02182-f002:**
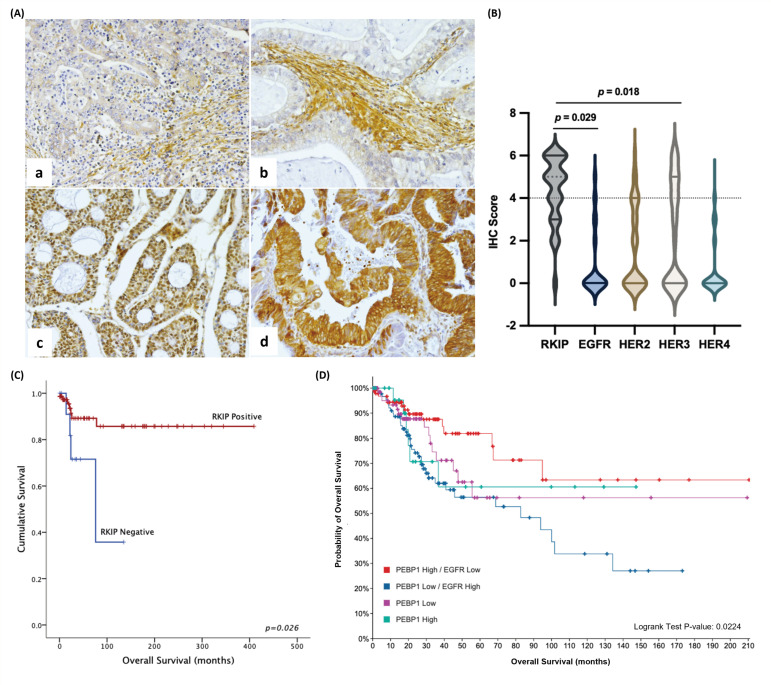
Correlation between RKIP and HER receptor expression in cervical cancer. (**A**) Immunohistochemistry analysis for RKIP in representative adenocarcinoma tissues of cervical cancer: (**a**) sample with low RKIP expression in the tumor and positive staining in the stroma; (**b**) sample showing positive staining in the stroma but with negative RKIP expression in the tumor; (**c**) sample with high nuclear expression of RKIP; (**d**) sample showing high cytoplasmatic RKIP expression. All pictures were taken at 200× magnification. (**B**) Violin plot showing the distribution of IHC scores for RKIP and HER receptors. For correlation analysis between RKIP and HER receptors, we considered the cases with IHC scores > 4: 100/181 for RKIP, 7/176 for EGFR, 23/170 for HER2, 70/166 for HER3, 1/158 for HER4. (**C**) Kaplan–Meier survival analysis was conducted with positive (red) and negative (blue) RKIP protein expression and its association with overall survival in cervical adenocarcinoma (N = 90). (**D**) Kaplan–Meier survival analysis was performed at cBioPortal considering patient tumor samples with both RKIP-low and EGFR-high (PEBP1: EXP < 0 and EGFR: EXP > 0; N = 99) or both RKIP-high and EGFR-low (PEBP1: EXP > 0 and EGFR: EXP < 0; N = 107) mRNA expression and patients with only RKIP-low (PEBP1: EXP < 0; N = 71) or RKIP-high (PEBP1: EXP > 0; N = 26) expression. For survival analysis, the log-rank test (*p* < 0.05) was considered statistically significant.

**Figure 3 cancers-16-02182-f003:**
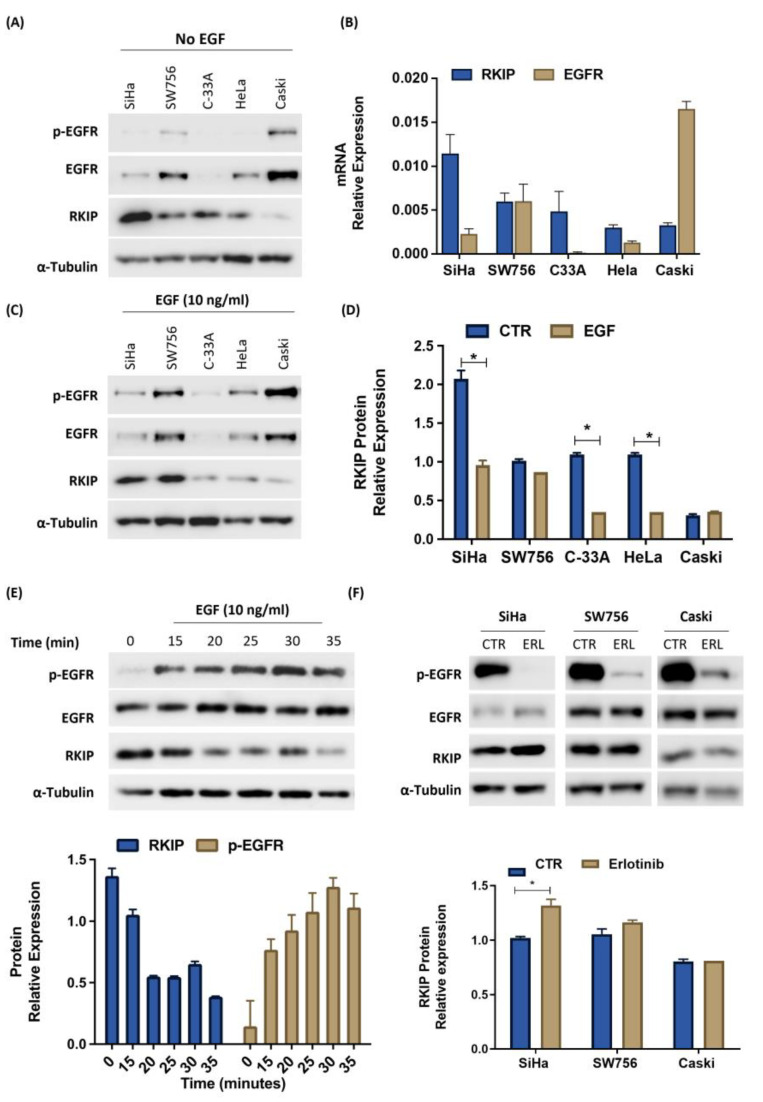
Effect of EGFR on RKIP expression. (**A**) Analysis of p-EGFR (Tyr1068), EGFR, and RKIP protein expression levels was performed using Western blotting. (**B**) Analysis of EGFR and RKIP mRNA expression levels was performed by qPCR. The results were calibrated to β-actin, which was used as a reference gene (N = 3). (**C**) Cells were stimulated with 10 ng/mL EGF for 15 min. The assay, run, and the blot revelation were performed simultaneously with blot A. (**D**) Western blot quantification for RKIP relative expression. (**E**) SW756 cells were stimulated with 10 ng/mL EGF for 15, 20, 25, 30, and 35 min. The isolated protein was separated by Western blotting, which was quantified for RKIP and p-EGFR. (**F**) The cell lines were treated for 2 h with erlotinib (ER), at 2.5 µM, followed by 15 min of treatment with 10 ng/mL EGF. The blots were quantified below for RKIP and p-EGFR. The Western blots shown are representative of at least two independent assays and were quantified using band densitometry analysis with ImageJ software version 1.8. Relative protein expression for RKIP was calculated as the ratio with α-tubulin and for p-EGFR as a ratio with total EGFR. The results are shown as the mean values obtained after quantification. *p* < 0.05 was considered statistically significant (*).

**Figure 4 cancers-16-02182-f004:**
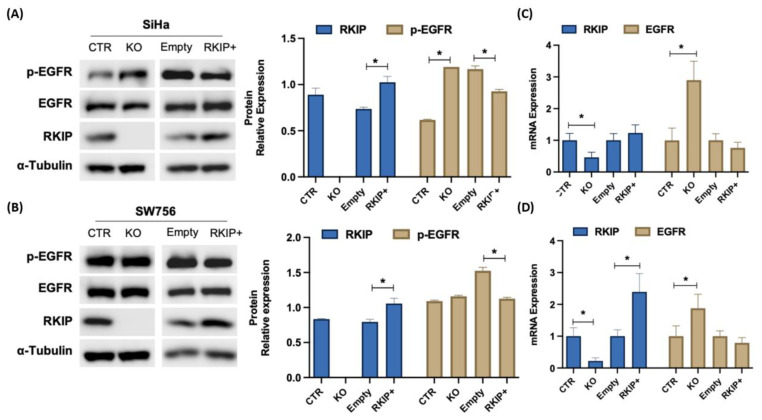
Does RKIP modulate EGFR expression? (**A**) Western blot analysis of p-EGFR (Tyr1068), EGFR, and RKIP protein expression in the SiHa cell line genetically manipulated to obtain RKIP KO and overexpression (RKIP+) under EGF-stimulating conditions (10 ng/mL, 15 min). (**B**) The same western blot analysis as A, but for SW756 cells. On the right side of the blot, band quantification was performed by band densitometry analysis using ImageJ software version 1.8. (**C**) Analysis of EGFR and RKIP mRNA expression levels by qPCR under the same conditions as in A. The results were calibrated to β-actin, which was used as a reference gene (N = 3). (**D**) The same qPCR analysis as in C, except for the SW756 cell line. Western blots are representative of at least 2 independent assays. Relative protein expression for RKIP was calculated as the ratio with α-tubulin and for p-EGFR as a ratio with total EGFR. The results are shown as the mean value achieved after quantification. *p* < 0.05 was considered statistically significant (*). CTR: CRISPR Control; KO: Knockout; Empty: pcDNA vector; RKIP+: pcDNA vector containing full RKIP cDNA.

**Figure 5 cancers-16-02182-f005:**
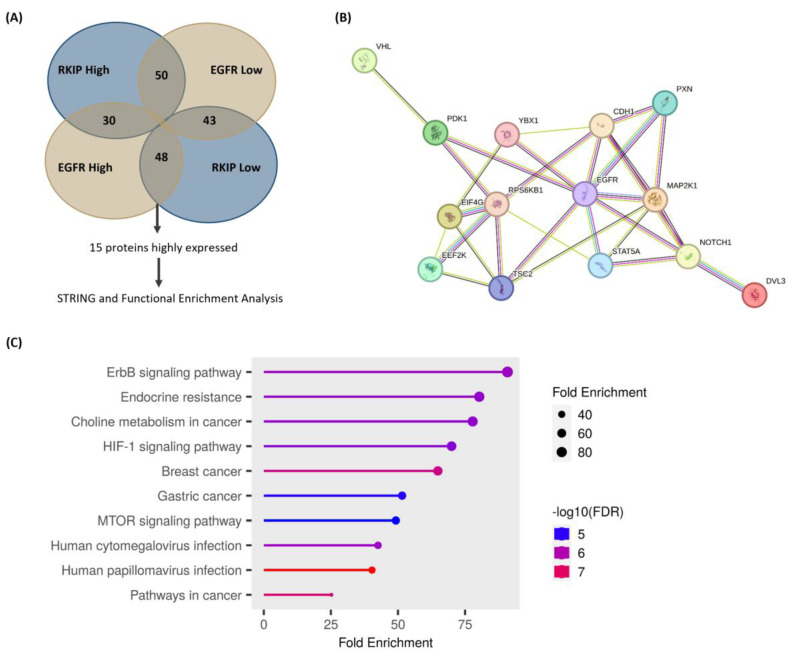
In silico analysis of protein expression signatures from cervical cancer patients with RKIP downregulation and EGFR upregulation. (**A**) Tumor samples from 48 patients with low RKIP mRNA expression (log RNA Seq V2 RSEM) and high EGFR protein expression (RPPA) were selected in cBioPortal using TCGA CSCC (Firehose legacy). Through enrichment analysis, 15 proteins were found to be highly expressed in this group of patients compared with the following groups of patients: RKIP-low mRNA expression and EGFR-high protein expression (PEBP1:EXP < 0 and EGFR:PROT > 0; N = 48); RKIP-high mRNA expression and EGFR-high protein expression (PEBP1:EXP > 0 and EGFR:PROT > 0; N = 30); RKIP-high mRNA expression and EGFR-low protein expression (PEBP1:EXP > 0 and EGFR:PROT < 0; N = 50); and RKIP-low mRNA and EGFR-low protein expression (PEBP1:EXP < 0 and EGFR:PROT < 0; N = 43). (**B**) Functional protein association network performed in STRING (https://string-db.org/, last accessed on 14 April 2024), showing a significant interaction within the network (PPI enrichment *p*-value = 2.22 × 10^−16^). (**C**) ShinyGo Lollipop plot results from the functional enrichment analysis showing the top 10 KEEG pathways enriched in the subset of enriched proteins. Enriched pathways were sorted considering the Fold Enrichment; the size of the circles is proportional to the Fold Enrichment, and the color of the bars corresponds to the −log10(FDR) values.

**Table 1 cancers-16-02182-t001:** Expression of RKIP and HER receptors in 30 PanCancer studies on solid tumors (10,719 samples) ^1^.

Mutual Exclusivity									Co-Expression	
A	B	Neither	A Not B	B Not A	Both	Log2 OR	*p*-Value *	Tendency	Spearman Correlation	*p*-Value *
PEBP1	EGFR	8480	510	835	25	−1.006	<0.001	Mutual exclusivity	−0.32	7.18 × 10^−229^
PEBP1	ERBB4	8972	499	343	36	0.916	<0.001	Co-occurrence	0.04	1.099 × 10^−5^
PEBP1	ERBB2	8476	469	839	66	0.508	0.008	Co-occurrence	0.06	9.32 × 10^−10^
PEBP1	ERBB3	8792	491	523	44	0.591	0.010	Co-occurrence	0.07	1.09 × 10^−10^

^1^ Data from mRNA expression z-scores relative to diploid samples (RNA Seq V2 RSEM) are available at cBioPortal (https://www.cbioportal.org, last accessed on 14 April 2024); OR: Odds Ratio; * Significance: *p* < 0.05.

**Table 2 cancers-16-02182-t002:** Top overexpressed proteins in the set of samples with RKIP mRNA downregulation and EGFR protein upregulation in cervical cancer (N = 48) ^1^.

Protein	Cytoband	*p*-Value	q-Value *	Molecular Function ^2^
**EGFR**	7p11.2	0.00	0.00	RTK signaling
**NOTCH1**	9q34.3	1.05 × 10^−13^	1.06 × 10^−11^	NOTCH signaling
**EEF2K**	16p12.2	1.504 × 10^−6^	8.466 × 10^−5^	Protein synthesis
**VHL**	3p25.3	1.668 × 10^−6^	8.466 × 10^−5^	HIF-1 signaling
**EIF4G1**	3q27.1	3.610 × 10^−6^	1.466 × 10^−4^	RNA binding and transport
**p-EGFR (Tyr1068)**		6.903 × 10^−6^	2.335 × 10^−4^	RTK signaling
**DVL3**	3q27.1	9.316 × 10^−5^	1.576 × 10^−3^	Wnt signaling
**STAT5A**	17q21.2	2.484 × 10^−4^	3.843 × 10^−3^	JAK-STAT signaling pathway
**MAP2K1**	15q22.31	2.650 × 10^−4^	3.843 × 10^−3^	MAPK signaling
**YBX1**	1p34.2	7.978 × 10^−4^	9.527 × 10^−3^	mRNA splicing
**p-PDK1 (Ser241)**		9.093 × 10^−4^	0.0103	HIF-1 signaling
**RPS6KB1**	17q23.1	1.719 × 10^−3^	0.0174	mTOR and PI3K/AKT signaling
**TSC2**	16p13.3	2.189 × 10^−3^	0.0212	mTOR and PI3K/AKT signaling
**PXN**	12q24.23	2.521 × 10^−3^	0.0213	Cell junction/ECM organization
**CDH1**	16q22.1	2.800 × 10^−3^	0.0219	Cell junction/ECM organization

^1^ Data from CbioPortal (https://www.cbioportal.org/, last accessed on 14 April 2024); * Significance: *p*- and q-value < 0.05; ^2^ STRING functional analysis (https://string-db.org/, last accessed on 14 April 2024).

## Data Availability

Data sharing is not applicable to this article.

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
