# Peer review of "Unveiling the RKIP and EGFR Inverse Relationship in Solid Tumors: A Case Study in Cervical Cancer"

_cancers, 2024, doi:10.3390/cancers16122182_

Round 1

Reviewer 1 Report

Comments and Suggestions for Authors

The authors presented the article entitled” Unveiling the RKIP and EGFR inverse relationship in solid tumors: A case study in cervical cancer”. The study method detailed, results comprehensive. In discussion section, RKIP and EGFR negative correlation in cervical cancer, how the relationship exists in other solid cancer such as lung cancer and colorectal cancer should be discussed.

Comments on the Quality of English Language

Minor editing of English language required

Author Response

We appreciate the reviewer's comments. All correlations in the 30 different tumor types analyzed are detailed in the manuscript and supplementary tables for readers to explore. However, further discussion of this RKIP/EGFR relationship in other tumors requires functional validation, so we believe that by doing so, we will be overestimating our findings.

The paper was reviewed by a native English speaker and teacher, so we believe that we have significantly improved the paper writing.

Reviewer 2 Report

Comments and Suggestions for Authors

This study by Cardoso-Carneiro et al. examines the correlation between RKIP and EGFR expression in solid tumours. The authors identify an inverse relationship that is shared across the majority of solid tumours (in contrast to the relationship between RKIP and other ErbB family members). Focussing their analyses on cervical cancer, they confirm this inverse relationship in a panel of cell lines and then assess the impact of either silencing or overexpressing RKIP. Although the correlation between EGFR and RKIP expression is somewhat interesting, I find this study limited in scope and lacking key experiments. As such, I struggle to see how the study in its current form significantly adds to our knowledge.

My issues include:

1.      The inverse correlation between RKIP and EGFR expression in the panel of 5 cervical cancer cell lines is not very convincing. For example, C33A cells have almost imperceptible levels of EGFR but only intermediate levels of RKIP. Further cell lines (of which there are many) could be added to this panel to strengthen the data.

2.      As the study focusses on cervical cancer, it would be interesting to determine the effect of HPV (which is the causative agent in ~98% of cervical cancers) on RKIP expression. Do the HPV oncoproteins E6 and E7 repress RKIP expression? The ERK-MAPK pathway is of critical importance during HPV infection and HPV-driven cancer, so if this was a novel mechanism by which HPV modulates the ERK-MAPK pathway it would be of significant interest to the field.

3.      There is no attempt to provide any mechanistic into how EGFR and RKIP expression are related. As the authors will be aware, a correlation does not necessarily indicate causation and the experiments in Fig 3D-F to not go far enough to address this. Is this a feed-forward loop whereby high MAPK activity represses RKIP expression? Does inhibiting MEK1/2 or ERK1/2 prevent the EGF-mediated suppression of RKIP? The data in Supp Fig 3B is interesting but further evidence would be required to confirm post-transcriptional regulation of RKIP (e.g. does proteosome inhibition with MG132 prevent EGF-induced loss of RKIP expression?).

4.      The authors assess protein levels of EGFR by western blot, but it is important to confirm the surface expression levels by e.g. immunofluorescence or flow cytometry.

5.      In Table 1 the authors claim that RKIP is “co-expressed with the other HER receptors”, but Spearman coefficients of 0.04, 0.06 and 0.07 indicate almost no correlation to me.

6.      Some of the inconsistencies seen between cell lines are not fully interrogated. Why does RKIP KO increase p-EGFR in SiHa but not SW756?

7.      In a previous study by the authors (https://pubmed.ncbi.nlm.nih.gov/23527098/) they perform a similar analysis, identifying that RKIP expression is low in cervical cancer. In it, they conclude that RKIP knockdown has no impact on EFGR/MAPK pathway activation in HeLa, SiHa and C33A cells, yet in the current study it is shown that RKIP knockout DOES increase EGFR mRNA expression and EGFR protein phosphorylation (at least in SiHa cells). What is the reason for this inconsistency?

8.      In the paper referenced in point 5, the authors are able to clearly detect EGFR protein by western blot in C33A cells. Why does expression appear to be so low in this cell line in this manuscript?

9.      Some of the western blot data presented (it is not clear which) is the result of only two biological repeats and therefore conclusions should not be drawn from these experiments, nor statistical analyses performed.

10.   The authors switch between RKIP and PEBP1 regularly which is confusing. They should stick to PEBP1 when referring to the gene/transcript and RKIP when referring to the protein.

Comments on the Quality of English Language

/

Author Response

Dear Reviewer

We appreciate the opportunity to clarify our manuscript and the editor’s and reviewer's insightful comments. The manuscript has been reviewed following the recommendations, as it has been improved to be clearer to the reader and/or to meet the journal submission guidelines. Please find below a detailed ‘point-by-point’ response to the reviewers’ comments.

This study by Cardoso-Carneiro et al. examines the correlation between RKIP and EGFR expression in solid tumours. The authors identify an inverse relationship that is shared across the majority of solid tumours (in contrast to the relationship between RKIP and other ErbB family members). Focussing their analyses on cervical cancer, they confirm this inverse relationship in a panel of cell lines and then assess the impact of either silencing or overexpressing RKIP. Although the correlation between EGFR and RKIP expression is somewhat interesting, I find this study limited in scope and lacking key experiments. As such, I struggle to see how the study in its current form significantly adds to our knowledge.

My issues include:

1. The inverse correlation between RKIP and EGFR expression in the panel of 5 cervical cancer cell lines is not very convincing. For example, C33A cells have almost imperceptible levels of EGFR but only intermediate levels of RKIP. Further cell lines (of which there are many) could be added to this panel to strengthen the data.

R: We appreciate and agree with the reviewer's comment. However, concerning the correlation between RKIP and EGFR in CC cell lines basal conditions, we were constating only a tendency and not taking further conclusions from Figure 3A. More important than looking at the basal conditions, when the cells were stimulated with EGF (Figure 3C-D), excepting SW756 and Caksi as explained in the manuscript, all of them (including C33A) depicted a concomitant decrease in RKIP levels, indicating that RKIP could be in fact under EGFR control. Further experiments with EGF stimulation over time and Erlotinib treatment corroborated the findings.

Regarding the number of cell lines and, compared with the number of commercially available cervical cancer cell lines we considered 5 a good number for those first comparisons (Figure 3)

2. As the study focuses on cervical cancer, it would be interesting to determine the effect of HPV (which is the causative agent in ~98% of cervical cancers) on RKIP expression. Do the HPV oncoproteins E6 and E7 repress RKIP expression? The ERK-MAPK pathway is of critical importance during HPV infection and HPV-driven cancer, so if this was a novel mechanism by which HPV modulates the ERK-MAPK pathway it would be of significant interest to the field.

R: We completely agree with the reviewer's point. Indeed, given that cervical cancer exhibits a notable percentage of RKIP loss, even in pre-cancer lesions, we have another manuscript under preparation that aims to understand the role of HPV infection in RKIP expression and the other ways around. Due to scientific secrecy, we cannot reveal the results, but we can anticipate that they are interesting. As the focus of the present manuscript is the correlation of EGFR with RKIP in solid tumors, we consider that including HPV results would completely deviate from the objective of the work even though in the functional/experimental part of the article we used CC as a model.

3. There is no attempt to provide any mechanistic into how EGFR and RKIP expression are related. As the authors will be aware, a correlation does not necessarily indicate causation and the experiments in Fig 3D-F to not go far enough to address this. Is this a feed-forward loop whereby high MAPK activity represses RKIP expression? Does inhibiting MEK1/2 or ERK1/2 prevent the EGF-mediated suppression of RKIP? The data in Supp Fig 3B is interesting but further evidence would be required to confirm post-transcriptional regulation of RKIP (e.g. does proteosome inhibition with MG132 prevent EGF-induced loss of RKIP expression?).

R: We appreciate the reviewer's suggestions, and recognize that we did not explore the putative intracellular mediators of this correlation. The reasons behind this were two:

-First, the aim of the work is based on a factual inverse correlation between RKIP and EGFR in the vast majority of solid tumors which may constitute a prognostic factor for patients. The choice to validate this correlation in CC cell lines is explained in the article, and this validation served to confirm that there is a mutual regulation, putatively in a negative feedback loop, between the two proteins and that it is not just a mere statistical correlation observed in silico and human samples. Thus, sticking to our major aim, modulating genetically and/or pharmacologically both molecules, we are completely convinced that we provide adequate information to establish a relation of causality underlying this correlation.

-Secondly, it is well-known and factual that RTKs, and especially EGFR, can shape the MAPK pathway through multiple secondary mediators. Likewise, despite its name, RKIP is a protein that modulates a wide range of intracellular pathways and not just the MAPK pathway (Reference 18). This means that if we want to know the secondary mechanisms underlying the EGFR and RKIP correlation, there is a long way to go, and it doesn't just go through the MPAK pathway. Furthermore, even if we wanted to explore the MAPK pathway, we know from the group's previous work that RKIP has no major impact on activating this pathway in CC (Reference 20).

4. The authors assess protein levels of EGFR by western blot, but it is important to confirm the surface expression levels by e.g. immunofluorescence or flow cytometry.

R: We appreciate the reviewer's comment. In the present paper, we recurred to western blot for being a more quantitative methodology, allowing us to establish correlations. However, we had already evaluated the EGFR immunocytochemistry analysis by IF in these cell lines, showing its main localization in cell membrane (Reference 8).

5. In Table 1 the authors claim that RKIP is “co-expressed with the other HER receptors”, but Spearman coefficients of 0.04, 0.06 and 0.07 indicate almost no correlation to me.

R: We agree with the reviewer's comment. It was a matter of a bad choice of vocabulary on our side. Thus, we changed the sentences regarding Table 1 as follows: “At the mRNA level, we found that PEBP1 is mutually exclusively expressed with EGFR and co-expressed with the other HER receptors (Table 1). Accordingly, by the co-expression plots, we observed that PEBP1 expression is significantly inversely correlated with EGFR expression (Spearman coefficient = -0.32), but none with the other HER receptors (Table 1).”

 6. Some of the inconsistencies seen between cell lines are not fully interrogated. Why does RKIP KO increase p-EGFR in SiHa but not SW756?

R: We appreciate the reviewer's comment. In the results section we elaborated on this topic by commenting that “RKIP depletion resulted in a significant increase in EGFR mRNA levels and induces its phosphorylation, more impactfully in cell lines with low basal activation levels of EGFR and without MAPK alterations (SiHa).” As we showed in this work, SW756 was the cell line with higher levels of EGFR and pEGFR. Consequently, higher doses of Erlotinib were needed to inhibit EGFR activation and, subsequently, increase RKIP expression. In fact, due to the known presence of KRAS G12C mutation, SW756 is highly addicted to EGFR/MAPK signaling. Hence modulating RKIP expression may not be enough to overcome the oncogene addiction state of this cell line.

7. In a previous study by the authors (https://pubmed.ncbi.nlm.nih.gov/23527098/) they perform a similar analysis, identifying that RKIP expression is low in cervical cancer. In it, they conclude that RKIP knockdown has no impact on EFGR/MAPK pathway activation in HeLa, SiHa and C33A cells, yet in the current study it is shown that RKIP knockout DOES increase EGFR mRNA expression and EGFR protein phosphorylation (at least in SiHa cells). What is the reason for this inconsistency?

R: Thank you for reading our previous work. We have to disagree that the results are inconsistent since the approach here was completely different. In the previous study, RKIP was knocked down using a short-hairpin approach resulting in only a reduction of RKIP expression, while in this study we completely abolish its expression by using CRISP-Cas9 technology. Also, the assays are not comparable since it was not done in the same way:

-In the first paper the cells, including Siha, were knockdown for RKIP and the levels of phospho-EGFR were evaluated before and after EGF stimulation. The samples were run together, hampering the evaluation of RKIP knockdown effects in p-EGFR basal levels expression. That’s why in the present submission the assay with RKIP knockout cells (Figure 4) was done at basal culture conditions and without EGF stimulation, which allowed us to verify that RKIP KO per se induces EGFR activation, a difference that clearly is not possible to see when the cells are simultaneously knockdown for RKIP and stimulated with EGF, as it was done in the first PlosOne paper. Moreover, even though an impact of RKIP KD on EGFR activation was not possible to detect, is clear to us that RKIP expression decreases in all cells knocked down for RKIP when stimulated with EGF, which even is an observation not mentioned in the first paper, is following what we observed in the 2024 assays.

- In the present paper, to corroborate the results we made a step forward by overexpressing RKIP in the same cell lines that were knockout, allowing us to demonstrate that the finding of p-EGFR be regulated by the levels of RKIP is consistent.

8. In the paper referenced in point 5, the authors are able to clearly detect EGFR protein by western blot in C33A cells. Why does expression appear to be so low in this cell line in this manuscript?

R: We don’t know exactly what is the reviewer referring to with “point 5”, but we consider that the reported EGFR expression levels in C33A, as in the other cell lines, by us in two previous works (References 8 and 20) are completely following the ones showed in present manuscript. We have obviously to be aware that the western blots are different, revealed in different apparatus, with different exposes and reagents which hampered the exact levels of expression in the 3 papers for each cell line, but the different expression levels between cell lines are exactly the same in the 3 articles.

9. Some of the western blot data presented (it is not clear which) is the result of only two biological repeats and therefore conclusions should not be drawn from these experiments, nor statistical analyses performed.

Regarding the N of the western blots, we were honest in saying that the quantification resulted from repeating the western blots at least twice, as in the case of Figure 3A and C, which are blots that we have already carried out in previous articles and we are sure of the consistency of the results. In the remaining assays, there were 3, 4 or more repetitions, the repetitions necessary to validate the results we were observing. The western blots were quantified by at least 3 researchers independently and it was the result of this quantification that allowed us to do the statistics even in the cases where the WB was done twice (which in itself was consistent removing the need for more repetitions). To make it clear, this information was added in the Materials and Methods section.

 10. The authors switch between RKIP and PEBP1 regularly which is confusing. They should stick to PEBP1 when referring to the gene/transcript and RKIP when referring to the protein.

R: We are aware that switching from the two names could be confusing, however, in cancer research the protein is well-known as RKIP and barely as PEBP1 (gene name), that’s why we maintained this taxonomy. Yet, we reviewed the paper to guarantee we refer to PEBP1 only when citing the gene.

Reviewer 3 Report

Comments and Suggestions for Authors

The authors realised a very interesting paper focused on the association between Raf Kinase Inhibitors Proteins and EGFR in different solid tumors with special interest for cervical cancer. The article reveals new insights regarding this issue. However in order to increase the strength of the manuscript we recommend adding more recent references.

Comments on the Quality of English Language

Minor English corrections are needed

Author Response

We appreciate the reviewer's suggestion. We agree that the number of references may seem small, however, we consider that for the article in question, the references cited are the most appropriate and necessary.

The paper was reviewed by a native English speaker and teacher, so we believe that we have significantly improved the paper writing.

Round 2

Reviewer 2 Report

Comments and Suggestions for Authors

I appreciate the corrections and clarifications provided by the authors to address my previous concerns.